# Synthesizing the preferred inputs for neurons in neural networks via deep generator networks

**Anh Nguyen**
anguyen8@uwyo.edu

**Alexey Dosovitskiy**
dosovits@cs.uni-freiburg.de

**Jason Yosinski**
jason@geometric.ai

**Thomas Brox**
brox@cs.uni-freiburg.de

**Jeff Clune**
jeffclune@uwyo.edu

## Abstract

Deep neural networks (DNNs) have demonstrated state-of-the-art results on many pattern recognition tasks, especially vision classification problems. Understanding the inner workings of such computational brains is both fascinating basic science that is interesting in its own right—similar to why we study the human brain—and will enable researchers to further improve DNNs. One path to understanding how a neural network functions internally is to study what each of its neurons has learned to detect. One such method is called *activation maximization* (AM), which synthesizes an input (e.g. an image) that highly activates a neuron. Here we dramatically improve the qualitative state of the art of activation maximization by harnessing a powerful, learned prior: a deep generator network (DGN). The algorithm (1) generates qualitatively state-of-the-art synthetic images that look almost real, (2) reveals the features learned by each neuron in an interpretable way, (3) generalizes well to new datasets and somewhat well to different network architectures without requiring the prior to be relearned, and (4) can be considered as a high-quality generative method (in this case, by generating novel, creative, interesting, recognizable images).

## 1   Introduction and Related Work

Understanding how the human brain works has been a long-standing quest in human history. Neuro-scientists have discovered neurons in human brains that selectively fire in response to specific, abstract concepts such as Halle Berry or Bill Clinton, shedding light on the question of whether learned neural codes are local vs. distributed [1]. These neurons were identified by finding the *preferred stimuli* (here, images) that highly excite a specific neuron, which was accomplished by showing subjects many different images while recording a target neuron's activation. Such neurons are multifaceted: for example, the "Halle Berry neuron" responds to very different stimuli related to the actress—from pictures of her face, to pictures of her in costume, to the word "Halle Berry" printed as text [1].

Inspired by such neuroscience research, we are interested in shedding light into the inner workings of DNNs by finding the preferred inputs for each of their neurons. As the neuroscientists did, one could simply show the network a large set of images and record a set of images that highly activate a neuron [2]. However, that method has disadvantages vs. *synthesizing* preferred stimuli: 1) it requires a distribution of images that are similar to those used to train the network, which may not be known (e.g. when probing a trained network when one does not know which data were used to train it); 2) even in such a dataset, many informative images that would activate the neuron may not exist because the image space is vast [3]; 3) with real images, it is unclear which of their features a neuron has learned: for example, if a neuron is activated by a picture of a lawn mower on grass, it is unclear if it

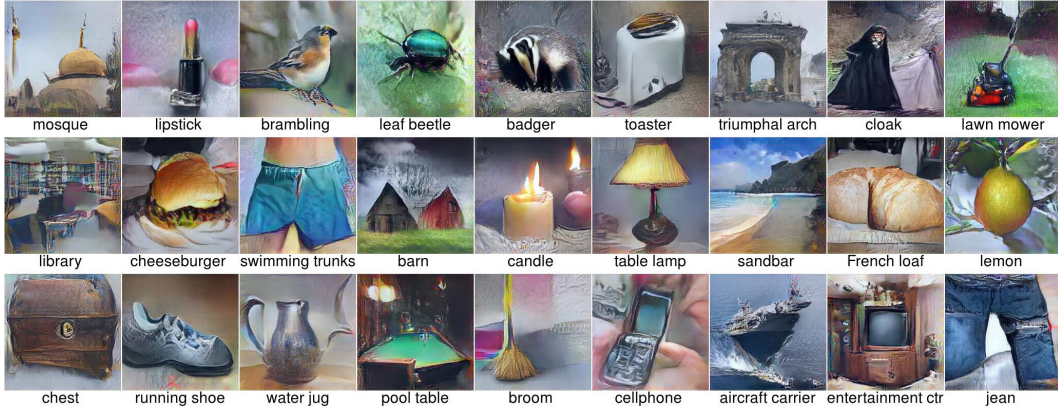

Figure 1: Images synthesized from scratch to highly activate output neurons in the CaffeNet deep neural network, which has learned to classify different types of ImageNet images.

'cares about' the grass, but if an image synthesized to highly activate the lawn mower neuron contains grass (as in Fig. 1), we can be more confident the neuron has learned to pay attention to that context.

Synthesizing preferred stimuli is called *activation maximization* [4–8, 3, 9]. It starts from a random image and iteratively calculates via backpropagation how the color of each pixel in the image should be changed to increase the activation of a neuron. Previous studies have shown that doing so without biasing the images produced creates unrealistic, uninterpretable images [5, 3], because the set of all possible images is so vast that it is possible to produce 'fooling' images that excite a neuron, but do not resemble the natural images that neuron has learned to detect. Instead, we must constrain optimization to generate only synthetic images that resemble natural images [6]. Attempting that is accomplished by incorporating *natural image priors* into the objective function, which has been shown to substantially improve the recognizability of the images generated [7, 6, 9]. Many hand-designed natural image priors have been experimentally shown to improve image quality such as: Gaussian blur [7], $\alpha$-norm [5, 7, 8], total variation [6, 9], jitter [10, 6, 9], data-driven patch priors [8], center-bias regularization [9], and initializing from mean images [9]. Instead of hand-designing such priors, in this paper, we propose to use a superior, *learned* natural image prior [11] akin to a generative model of images. This prior allows us to synthesize highly human-interpretable preferred stimuli, giving additional insight into the inner functioning of networks. While there is no way to rigorously measure human-interpretability, a problem that also makes quantitatively assessing generative models near-impossible [12], we should not cease scientific work on improving qualitative results simply because humans must subjectively evaluate them.

Learning generative models of natural images has been a long-standing goal in machine learning [13]. Many types of neural network models exist, including probabilistic [13], auto-encoder [13], stochastic [14] and recurrent networks [13]. However, they are typically limited to relatively low-dimensional images and narrowly focused datasets. Recently, advances in network architectures and training methods enabled the generation of high-dimensional realistic images [15, 16, 11]. Most of these works are based on Generative Adversarial Networks (GAN) [17], which trains two models simultaneously: a generative model $G$ to capture the data distribution, and a discriminative model $D$ to estimates the probability that a sample came from the training data rather than $G$. The training objective for $G$ is to maximize the probability of $D$ making a mistake. Recently Dosovitskiy and Brox [11] trained networks capable of generating images from highly compressed feature representations, by combining an auto-encoder-style approach with GAN's adversarial training. We harness these image *generator* networks as priors to produce synthetic preferred images. These generator networks are close to, but not true, generative models because they are trained without imposing any prior on the hidden distribution as in variational auto-encoders [14] or GANs [17], and without the addition of noise as in denoising auto-encoders [18]. Thus, there is no natural sampling procedure nor an implicit density function over the data space.

The image *generator* DNN that we use as a prior is trained to take in a code (e.g. vector of scalars) and output a synthetic image that looks as close to real images from the ImageNet dataset [19] as possible. To produce a preferred input for a neuron in a given DNN that we want to visualize, we optimize in the input code space of the image generator DNN so that it outputs an image that activates

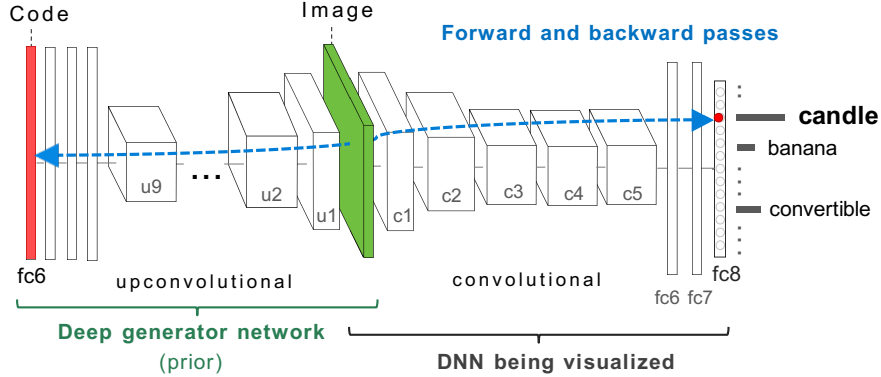

Figure 2: To synthesize a preferred input for a target neuron $h$ (e.g. the "candle" class output neuron), we optimize the hidden code input (red bar) of a deep image generator network (DGN) to produce an image that highly activates $h$. In the example shown, the DGN is a network trained to invert the feature representations of layer fc6 of CaffeNet. The target DNN being visualized can be a different network (with a different architecture and or trained on different data). The gradient information (blue-dashed line) flows from the layer containing $h$ in the target DNN (here, layer fc8) all the way through the image back to the input code layer of the DGN. Note that both the DGN and target DNN being visualized have fixed parameters, and optimization only changes the DGN input code (red).

the neuron of interest (Fig. 2). Our method restricts the search to only the set of images that can be drawn by the prior, which provides a strong biases toward realistic visualizations. Because our algorithm uses a deep generator network to perform activation maximization, we call it DGN-AM.

## 2  Methods

**Networks that we visualize.** We demonstrate our visualization method on a variety of different networks. For reproducibility, we use pretrained models freely available in Caffe or the Caffe Model Zoo [20]: CaffeNet [20], GoogleNet [21], and ResNet [22]. They represent different convnet architectures trained on the ~1.3-million-image 2012 ImageNet dataset [23, 19]. Our default DNN is CaffeNet [20], a minor variant of the common AlexNet architecture [24] with similar performance [20]. The last three fully connected layers of the 8-layer CaffeNet are called fc6, fc7 and fc8 (Fig. 2). fc8 is the last layer (pre softmax) and has 1000 outputs, one for each ImageNet class.

**Image generator network.** We denote the DNN we want to visualize by $\Phi$. Instead of previous works, which directly optimized an image so that it highly activates a neuron $h$ in $\Phi$ and optionally satisfies hand-designed priors embedded in the cost function [5, 7, 9, 6], here we optimize in the input code of an image generator network $G$ such that $G$ outputs an image that highly activates $h$.

For $G$ we use networks made publicly available by [11] that have been trained with the principles of GANs [17] to reconstruct images from hidden-layer feature representations within CaffeNet [20]. How $G$ is trained includes important differences from the original GAN configuration [17]. Here we can only briefly summarize the training procedure; please see [11] for more details. The training process involves four convolutional networks: 1) a *fixed* encoder network $E$ to be inverted, 2) a generator network $G$, 3) a *fixed* "comparator" network $C$ and 4) a discriminator $D$. $G$ is trained to invert a feature representation extracted by the network $E$, and has to satisfy three objectives: 1) for a feature vector $\mathbf{y}_i = E(\mathbf{x}_i)$, the synthesized image $G(\mathbf{y}_i)$ has to be close to the original image $\mathbf{x}_i$; 2) the features of the output image $C(G(\mathbf{y}_i))$ have to be close to those of the real image $C(\mathbf{x}_i)$; 3) $D$ should be unable to distinguish $G(\mathbf{y}_i)$ from real images. The objective for $D$ is to discriminate between synthetic images $G(\mathbf{y}_i)$ and real images $\mathbf{x}_i$ as in the original GAN [17].

In this paper, the encoder $E$ is CaffeNet truncated at different layers. We denote CaffeNet truncated at layer $l$ by $E_l$, and the network trained to invert $E_l$ by $G_l$. The "comparator" $C$ is CaffeNet up to layer pool5. $D$ is a convolutional network with 5 convolutional and 2 fully connected layers. $G$ is an upconvolutional (aka deconvolutional) architecture [15] with 9 upconvolutional and 3 fully connected layers. Detailed architectures are provided in [11].

**Synthesizing the preferred images for a neuron.** Intuitively, we search in the input code space of the image generator model $G$ to find a code $\mathbf{y}$ such that $G(\mathbf{y})$ is an image that produces high activation of the target neuron $h$ in the DNN $\Phi$ that we want to visualize (i.e. optimization maximizes $\Phi_h(G(\mathbf{y}))$). Recall that $G_l$ is a generator network trained to reconstruct images from the $l$-th layer features of CaffeNet. Formally, and including a regularization term, we may pose the activation maximization problem as finding a code $\widehat{\mathbf{y}^l}$ such that:

$$\widehat{\mathbf{y}^l} = \arg\max_{\mathbf{y}^l}(\Phi_h(G_l(\mathbf{y}^l)) - \lambda\|\mathbf{y}^l\|) \tag{1}$$

Empirically, we found a small amount of $L_2$ regularization ($\lambda = 0.005$) works best. We also compute the activation range for each neuron in the set of codes $\{\mathbf{y}_i^l\}$ computed by running validation set images through $E_l$. We then clip each neuron in $\widehat{\mathbf{y}^l}$ to be within the activation range of $[0, 3\sigma]$, where $\sigma$ is one standard deviation around the mean activation (the activation is lower bounded at 0 due to the ReLU nonlinearities that exist at the layers whose codes we optimize). This clipping acts as a primitive prior on the code space and substantially improves the image quality. In future work, we plan to learn this prior via a GAN or other generative model. Because the true goal of activation maximization is to generate interpretable preferred stimuli for each neuron, we performed random search in the hyperparameter space consisting of $L_2$ weight $\lambda$, number of iterations, and learning rate. We chose the hyperparameter settings that produced the highest quality images. We note that we found no correlation between the activation of a neuron and the recognizability of its visualization. Our code and parameters are available at `http://EvolvingAI.org/synthesizing`.

## 3   Results

### 3.1   Comparison between priors trained to invert features from different layers

Since a generator model $G_l$ could be trained to invert feature representations of an arbitrary layer $l$ of $E$, we sampled $l = \{3, 5, 6, 7\}$ to explore the impact on this choice and identify qualitatively which produces the best images. Here, the DNN to visualize $\Phi$ is the same as the encoder $E$ (CaffeNet), but they can be different (as shown below). The $G_l$ networks are from [11]. For each $G_l$ network we chose the hyperparameter settings from a random sample that gave the best qualitative results.

Optimizing codes from the convolutional layers ($l = 3, 5$) typically yields highly repeated fragments, whereas optimizing fully-connected layer codes produces much more coherent global structure (Fig. S13). Interestingly, previous studies have shown that $G$ trained to invert lower-layer codes (smaller $l$) results in far better reconstructions than higher-layer codes [25, 6]. That can be explained because those low-level codes come from natural images, and contain more information about image details than more abstract, high-level codes. For activation maximization, however, we are *synthesizing* an entire layer code from scratch. We hypothesize that this process works worse for $G_l$ priors with smaller $l$ because each feature in low-level codes has a small, local receptive field. Optimization thus has to independently tune features throughout the image without knowing the global structure. For example, is it an image of one or four robins? Because fully-connected layers have information from all areas of the image, they represent information such as the number, location, size, etc. of an object, and thus all the pixels can be optimized toward this agreed upon structure. An orthogonal, non-mutually-exclusive hypothesis is that the code space at a convolutional layer is much more high-dimensional, making it harder to optimize.

We found that optimizing in the fc6 code space produces the best visualizations (Figs. 1 & S13). We thus use this $G_6$ DGN as the default prior for the experiments in the rest of the paper. In addition, our images qualitatively appear to be the most realistic-looking compared to visualizations from all previous methods (Fig. S17). Our result reveals that a great amount of fine detail and global structure are captured by the DNN even at the last output layer. This finding is in contrast to a previous hypothesis that DNNs trained with supervised learning often ignore an object's global structure, and only learn discriminative features per class (e.g. color or texture) [3]. Section 3.5 provides evidence that this global structure does not come from the prior.

To test whether our method memorizes the training set images, we retrieved the closest images from the training set for each of sample synthetic images. Specifically, for each synthetic image for an output neuron $Y$ (e.g. lipstick), we find an image among the same class $Y$ with the lowest Euclidean distance in pixel space, as done in previous works [17], but also in each of the 8 code spaces of the

encoder DNN. While this is a much harder test than comparing to a nearest neighbor found among the entire dataset, we found no evidence that our method memorizes the training set images (Fig. S22). We believe evaluating similarity in the spaces of deep representations, which better capture semantic aspects of images, is a more informative approach compared to evaluating only in the pixel space.

## 3.2 Does the learned prior trained on ImageNet generalize to other datasets?

We test whether the same DNN prior ($G_6$) that was trained on inverting the feature representations of ImageNet images generalizes to enable visualizing DNNs trained on different datasets. Specifically, we target the output neurons of two DNNs downloaded from Caffe Model Zoo [20]): (1) An AlexNet DNN that was trained on the 2.5-million-image MIT Places dataset to classify 205 types of places with 50.1% accuracy [26]. (2) A hybrid architecture of CaffeNet and the network in [2] created by [27] to classify actions in videos by processing each frame of the video separately. The dataset consists of 13,320 videos categorized into 101 human action classes.

For DNN 1, the prior trained on ImageNet images generalizes well to the completely different MIT Places dataset (Fig. 3). This result suggests the prior trained on ImageNet will generalize to other natural image datasets, at least if the architecture of the DNN to be visualized $\Phi$ is the same as the architecture of the encoder network $E$ from which the generator model $G$ was trained to invert feature representations. For DNN 2: the prior generalizes to produce decent results; however, the images are not qualitatively as sharp and clear as for DNN 1 (Fig. 4). We have two orthogonal hypotheses for why this happens: 1) $\Phi$ (the DNN from [27]) is a heavily modified version of $E$ (CaffeNet); 2) the two types of images are too different: the primarily object-centric ImageNet dataset vs. the UCF-101 dataset, which focuses on humans performing actions. Sec. 3.3 returns to the first hypothesis regarding how the similarity between $\Phi$ and $E$ affects the image quality

Overall, the prior trained with a CaffeNet encoder generalizes well to visualizing other DNNs of the same CaffeNet architecture trained on different datasets.

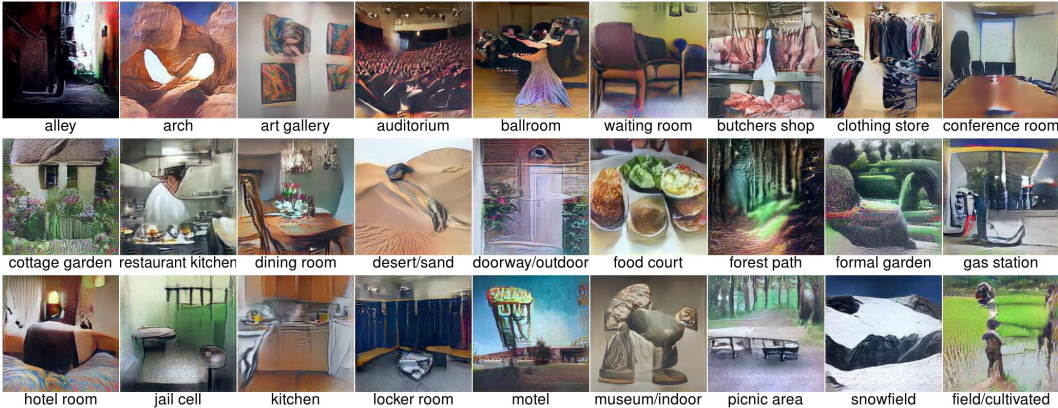

Figure 3: Preferred stimuli for output units of an AlexNet DNN trained on the MIT Places dataset [26], showing that the ImageNet-trained prior generalizes well to a dataset comprised of images of scenes.

## 3.3 Does the learned prior generalize to visualizing different architectures?

We have shown that when the DNN to be visualized $\Phi$ is the same as the encoder $E$, the resultant visualizations are quite realistic and recognizable (Sec. 3.1). To visualize a different network architecture $\hat{\Phi}$, one could train a new $\hat{G}$ to invert $\hat{\Phi}$ feature representations. However, training a new $G$ DGN for every DNN we want to visualize is computationally costly. Here, we test whether the same DGN prior trained on CaffeNet ($G_6$) can be used to visualize two state-of-the-art DNNs that are architecturally different from CaffeNet, but were trained on the same ImageNet dataset. Both were downloaded from Caffe Model Zoo and have similar accuracy scores: (a) GoogLeNet is a 22-layer network and has a top-5 accuracy of 88.9% [21]; (b) ResNet is a new type of very deep architecture with skip connections [22]. We visualize a 50-layer ResNet that has a top-5 accuracy of 93.3%. [22].

DGN-AM produces the best image quality when $\Phi = E$, and the visualization quality tends to degrade as the $\Phi$ architecture becomes more distant from $E$ (Fig. 5, top row; GoogLeNet is closer

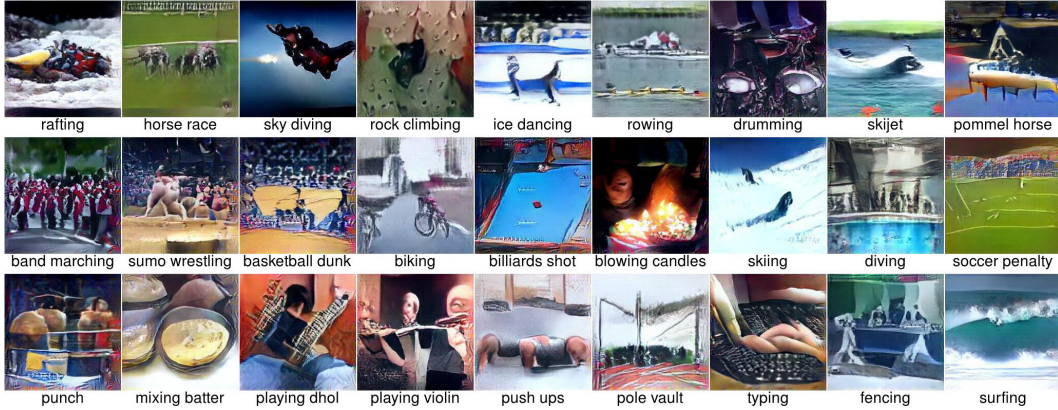

Figure 4: Preferred images for output units of a heavily modified version of the AlexNet architecture trained to classify videos into 101 classes of human activities [27]. Here, we optimize a single preferred image per neuron because the DNN only classifies single frames (whole video classification is done by averaging scores across all video frames).

in architecture to CaffeNet than ResNet) . An alternative hypothesis is that the network depth impairs gradient propagation during activation maximization. In any case, training a general prior for activation maximization that generalizes well to different network architectures, which would enable comparative analysis between networks, remains an important, open challenge.

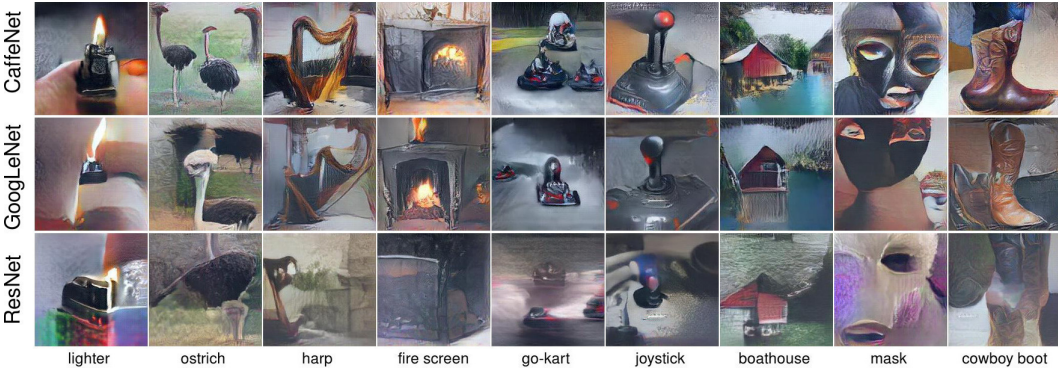

Figure 5: DGN-AM produces the best image quality when the DNN being visualized $\Phi$ is the same as the encoder $E$ (here, CaffeNet), as in the top row, and degrades when $\Phi$ is different from $E$.

### 3.4 Does the learned prior generalize to visualizing hidden neurons?

**Visualizing the hidden neurons in an ImageNet DNN.** Previous visualization techniques have shown that low-level neurons detect small, simple patterns such as corners and textures [2, 9, 7], mid-level neurons detect single objects like faces and chairs [9, 2, 28, 7], but that visualizations of hidden neurons in fully-connected layers are alien and difficult to interpret [9]. Since DGN was trained to invert the feature representations of real, full-sized ImageNet images, one possibility is that this prior may not generalize to producing preferred images for such hidden neurons because they are often smaller, different in theme, and or do not resemble real objects. To find out, we synthesized preferred images for the hidden neurons at all layers and compare them to images produced by the multifaceted feature visualization method from [9], which harnesses hand-designed priors of total variation and mean image initialization. The DNN being visualized is the same as in [9] (the CaffeNet architecture with weights from [7]).

The side-by-side comparison (Fig. S14) shows that both methods often agree on the features that a neuron has learned to detect. However, overall DGN-AM produces more realistic-looking color and texture, despite not requiring optimization to be seeded with averages of real images, thus improving our ability to learn what feature each hidden neuron has learned. An exception is for the faces of

human and other animals, which DGN-AM does not visualize well (Fig. S14, 3rd unit on layer 6; 1st unit on layer 5; and 6th unit on layer 4).

**Visualizing the hidden neurons in a Deep Scene DNN.** Recently, Zhou et al. [28] found that object detectors automatically emerge in the intermediate layers of a DNN as we train it to classify scene categories. To identify what a hidden neuron cares about in a given image, they densely slide an occluding patch across the image and record when activation drops. The activation changes are then aggregated to segment out the exact region that leads to the high neural activation (Fig. 6, the highlighted region in each image). To identify the semantics of these segmentations, humans are then shown a collection of segmented images for a specific neuron and asked to label what types of image features activate that neuron [28]. Here, we compare our method to theirs on an AlexNet DNN trained to classify 205 categories of scenes from the MIT Places dataset (described in Sec. 3.2).

The prior learned on ImageNet generalizes to visualizing the hidden neurons of a DNN trained on the MIT Places dataset (Fig. S15). Interestingly, our visualizations produce similar results to the method in [28] that requires showing each neuron a large, external dataset of images to discover what feature each neuron has learned to detect (Fig. 6). Sometimes, DGN-AM reveals additional information: a unit that fires for TV screens also fires for people on TV (Fig. 6, unit 106). Overall, DGN-AM thus not only generalizes well to a different dataset, but also produces visualizations that qualitatively fall within the human-provided categories of what type of image features each neuron responds to [28].

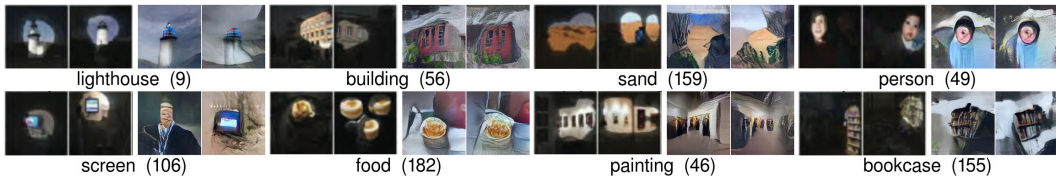

Figure 6: Visualizations of example hidden neurons at layer 5 of an AlexNet DNN trained to classify categories of scenes from [28]. For each unit: we compare the two visualizations produced by a method from [28] (left) to two visualizations produced by our method (right). The left two images are real images, each highlighting a region that highly activates the neuron, and humans provide text labels describing the common theme in the highlighted regions. Our synthetic images enable the same conclusion regarding what feature a hidden neuron has learned. An extended version of this figure with more units is in Fig. S16. Best viewed electronically with zoom.

### 3.5   Do the synthesized images teach us what the neurons prefer or what the prior prefers?

**Visualizing neurons trained on unseen, modified images.** We have shown that DGN-AM can generate preferred image stimuli with realistic colors and coherent global structures by harnessing the DGN's strong, learned, natural image prior (Fig. 1). To what extent do the global structure, natural colors, and sharp textures (e.g. of the brambling bird, Fig. 1) reflect the features learned by the "brambling" neuron vs. those preferred by the prior? To investigate that, we train 3 different DNNs: one on images that have less global structure, one on images of non-realistic colors, and one on blurry images. We test whether DGN-AM with *the same prior* produces visualizations that reflect these modified, unrealistic features.

Specifically, we train 3 different DNNs following CaffeNet architecture to discriminate 2000 classes. The first 1000 classes contain regular ImageNet images, and the 2nd 1000 classes contain *modified* ImageNet images. We perform 3 types of modifications: 1) we cut up each image into quarters and re-stitch them back in a random order (Fig. S19); 2) we convert regular RGB into BRG images (Fig. S20); 3) we blur out images with Gaussian blur with radius of 3 (Fig. S21).

We visualize both groups of output neurons (those trained on 1000 regular vs. 1000 modified classes) in each DNN (Figs. S19, S20, & S21). The visualizations for the neurons that are trained on regular images often show coherent global structures, realistic-looking colors and sharpness. In contrast, the visualizations for neurons that are trained on modified images indeed show cut-up objects (Fig. S19), images in BRG color space (Fig. S20), and objects with washed out details (Fig. S21). The results show that DGN-AM visualizations do closely reflect the features learned by neurons from the data and that these properties are not exclusively produced by the prior.

**Why do visualizations of some neurons not show canonical images?** While many DGN-AM visualizations show global structure (e.g. a single, centered table lamp, Fig. 1); some others do not (e.g. blobs of textures instead of a dog with 4 legs, Fig. S18) or otherwise are non-canonical (e.g. a school bus off to the side of an image, Fig. S7). Sec. S5 describes our experiments investigating whether this is a shortcoming of our method or whether these non-canonical visualizations reflect some property of the neurons. The results suggest that DGN-AM can accurately visualize a class of images if the images of that set are mostly canonical, and the reason why the visualizations for some neurons lack global structure or are not canonical is that the set of images that neuron has learned to detect are often diverse (multi-modal), instead of having canonical pose. More research is needed into multifaceted feature visualization algorithms that separately visualize each type of image that activates a neuron [9].

### 3.6  Other applications of our proposed method

DGN-AM can also be useful for a variety of other important tasks. We briefly describe our experiments for these tasks, and refer the reader to the supplementary section for more information.

1. One advantage of synthesizing preferred images is that we can watch how features evolve during training to better understand what occurs during deep learning. Doing so also tests whether the learned prior (trained to invert features from a well-trained encoder) generalizes to visualizing underfit and overfit networks. The results suggest that the visualization quality is indicative of a DNN's validation accuracy to some extent, and the learned prior is not overly specialized to the well-trained encoder DNN. See Sec. S6 for more details.

2. Our method for synthesizing preferred images could naturally be applied to synthesize *preferred videos* for an activity recognition DNN to better understand how it works. For example, we found that a state-of-the-art DNN classifies videos without paying attention to temporal information across video frames (Sec. S7).

3. Our method can be extended to produce creative, original art by synthesizing images that activate two neurons at the same time (Sec. S8).

## 4  Discussion and Conclusion

We have shown that activation maximization—synthesizing the preferred inputs for neurons in neural networks—via a learned prior in the form of a deep generator network is a fruitful approach. DGN-AM produces the most realistic-looking, and thus interpretable, preferred images to date, making it qualitatively the state of the art in activation maximization. The visualizations it synthesizes from scratch improve our ability to understand which features a neuron has learned to detect. Not only do the images closely reflect the features learned by a neuron, but they are visually interesting. We have explored a variety of ways that DGN-AM can help us understand trained DNNs. In future work, DGN-AM or its learned prior could dramatically improve our ability to synthesize an image from a text description of it (e.g. by synthesizing the image that activates a certain caption) or create more realistic "deep dream" [10] images. Additionally, that the prior used in this paper does not generalize equally well to DNNs of different architectures motivates research into how to train such a general prior. Successfully doing so could enable informative comparative analyses between the information transformations that occur within different types of DNNs.

#### Acknowledgments

The authors would like to thank Yoshua Bengio for helpful discussions and Bolei Zhou for providing images for our study. Jeff Clune was supported by an NSF CAREER award (CAREER: 1453549) and a hardware donation from the NVIDIA Corporation. Jason Yosinski was supported by the NASA Space Technology Research Fellowship and NSF grant 1527232. Alexey Dosovitskiy and Thomas Brox acknowledge funding by the ERC Starting Grant VideoLearn (279401).

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
