[Supplementary Material]

# Supplementary materials for:
# Synthesizing the preferred inputs for neurons in neural networks via deep generator networks

## S5 Why do visualizations of some neurons not show canonical images?

While the visualizations of many neurons appear to be great-looking, showing *canonical* images of a class (e.g. a table lamp with shade and base, Fig. 1); many others do not (e.g. dog visualizations often show blobs of texture instead of a dog standing with 4 legs, Fig. S18). We investigate whether this is a shortcoming of our method or these non-canonical visualizations are actually reflecting some property of the neurons.

In this experiment, we aim to visualize a DNN trained on pairs of classes, in which one contain canonical images, and the other do not, and see if the visualizations reflect these classes. Specifically, we take 5 classes $\{C_i\}$ for which we found the visualizations did not show canonical images: school bus, cup, irish terrier, tabby cat, and hartebeest, and move all canonical images in each class $C_i$ into a new class $C_i'$. Data augmentation is perform for each pair of $C_i$ and $C_i'$ so they all have $\sim$1300 images as other classes. We add these resultant 10 classes back to the ImageNet training set, and train a DNN to classify between all 1005 classes.

Our method indeed generates canonical visualizations for neurons trained on canonical images (Fig. S7, entire school bus with wheels and windows, irish terrier standing on feet, tabby in front standing pose). This result shows evidence that our method reflects well the features learned by the neurons. The result for neurons that are trained on non-canonical images appear similar to many non-canonical visualizations we found previously (Fig. S18). In fact, in the training set, each of these 5 classes only contain a small percentage of images that are canonical: school bus (2%), tabby cat (3%), irish terrier (4%), hartebeest (6%), and cup (18%). These numbers for classes for which our visualizations often show canonical images are often much higher: table lamp (31%), brambling (49%), lipstick (29%), joystick (19%) and beacon (39%) (see Fig. 1 for example visualizations of these classes).

In overal, the evidence suggests that our method reflects well the features learned by the neurons. It seems, the visualizations of some neurons do not show canonical images simply because the set of features these neuron have learned to detect are diverse, and not canonical.

## S6 Visualizing under-trained, well-trained, and overfit networks

Here we test how do the visualizations for under-trained, well-trained, and overfit networks look, and if the image quality reflects the generalization ability (in terms of validation accuracy) of a DNN. To do this, we train a CaffeNet DNN with the training hyperparameters provided by the Caffe framework. Then we run our method to visualize the preferred stimuli for output and hidden neurons of network snapshots taken every 10,000 iterations. The resultant two videos for this experiment are available for review at: `https://www.youtube.com/watch?v=q4yIwiYH6FQ` and `https://www.youtube.com/watch?v=G8AtatM1Sts`.

Our result shows that the accuracy of the DNN seems to correlate with the visualization quality during the first 200,000 iterations. The features appear blurry initially and evolve to be clearer and clearer as the accuracy increases. Our method can be used to learn more about how features are being learned. For example, looking at the set of images that activate the "swimming trunks" neuron in the video, it seems that the concept of "swimming trunks" was associated with people in a blue ocean background in early iterations and gradually changes to the actual clothing item at around 300,000 iterations.

We are also interested in finding out whether the image quality would appear better or worse if a DNN overfits to training data. To check this, we re-train two DNNs –one on only 10% and one on 25% of the original ∼1.3M ImageNet images – so that they have a top-1 training accuracy of 100%, and validation accuracy of 20.2% and 31.5% respectively. The visualizations of these two DNNs appear recognizable, but worse than those of a well-trained DNN that has a validation accuracy of 57.4% and a training accuracy of 79.5% (Fig. S9). All three DNNs are given the same number of training updates.

Overall, the evidence supports the hypothesis that the visualization quality does correlate with a DNN's validation accuracy. Note that this is not true for *class* accuracy where the visualizations of output neurons with lowest class accuracy scores are still beautiful and recognizable (Fig. S10), suggesting that low accuracy scores are the result of DNN being confused by pairs of very similar classes in the dataset (e.g. sunglass vs. sunglasses). The result also shows that our learned prior does not overfit too much to a well-trained encoder DNN because the visualizations for under-trained and overfit networks are still sensible.

## S7    Visualizing an activity recognition network

Our method for synthesizing the preferred images could naturally be applied to synthesize preferred *videos* (i.e. sequences of images) for an activity recognition DNN. Here, we synthesize videos for LRCN—an activity recognition model made available by Donahue et al. [27]. The model combines a convolutional neural network (for feature extraction from each frame), and a LSTM recurrent network [27]. It was trained to classify videos into 101 classes of human activities in the UCF-101 dataset.

We synthesize a video of 160 frames (that is, the same length as videos in the training set) for each of the output neurons. The resultant videos are available for review at: https://www.youtube.com/watch?v=IOYnIK6N5Bg. The videos appear qualitatively sensible, but not as great as our best images in the main text. An explanation for this is that the convolutional network in the LRCN model is not CaffeNet, but instead a hybrid of two different popular convnet architectures [27].

An interesting finding about the inner working of this specific activity recognition model is that it does not care about the frame order, explaining the non-smooth transition between frames in synthetic videos. In fact, as we tested, shuffling all the frames in a real video also does not substantially change the classification decision of the DNN. Researchers could use our tool to discover such property of a given DNN, and improve it if necessary. For example, one could re-train this LRCN DNN to make it learn the correct temporal consistency across the frames in real videos (e.g. the smooth transition between frames, the intro/ending styles, etc.).

## S8    Synthesizing creative art by activating two neurons instead of one

A natural extension of our method is synthesizing images that activate multiple neurons at the same time instead of one. We found that optimizing a code to activate two neurons at the same time by simply adding an additional objective for the second neuron often leads to one neuron dominating the search. For example, say the "bell pepper" neuron happens to be easier to activate than the "candle" neuron, the final image will be purely an image of a bell pepper, and there will be no candles.

Here, to produce interesting results, we experiment with encouraging two neurons to be similarly activated. Specifically, we add an additional $L_2$ penalty for the distance between the two activations. Formally, we may pose the activation maximization problem for activating two units $h1$ and $h2$ of a network $\Phi$ via an image generator model $G_l$ as finding a code $\widehat{\mathbf{y}^l}$ such that:

$$\widehat{\mathbf{y}^l} = \arg\max_{\mathbf{y}^l}(\Phi_{h1}(G_l(\mathbf{y}^l)) + \Phi_{h2}(G_l(\mathbf{y}^l)) - \lambda\|\mathbf{y}^l\| - \gamma\|\Phi_{h1}(G_l(\mathbf{y}^l)) - \Phi_{h2}(G_l(\mathbf{y}^l))\|) \quad (2)$$

where $\gamma$ is the weight of the additional penalty term.

We found that the resultant visualizations are very interesting and diverse, and vary depending on which pair of neurons is activated. The following cases are observed:

1. Two objects blending into one new sensible type of object as the color of one object is painted on the global structure of the other (Fig. S11, goldfish + brambling = red brambling, spider monkey + brambling = yellow spider monkey)

2. Two objects blending into one new unrealistic, but *artistically interesting* type of object (Fig. S11, gazelle + brambling = a brambling with gazelle horns; scuba diver + brambling = a brambling underwater)

3. Two objects blending into one coherent picture that contains both (Fig. S11, dining table + brambling = brambling sitting on a dining table; apron + brambling = an apron with a brambling image printed on).

4. Activating two neurons uncovering a visualization of a new, unique facet of either neuron. Examples from Fig. S12: combining "American lobster" and "candles" leads to an image of people eating lobster that is on a plate (instead of a lobster on fire); combining "prison" and "candles" results in an outside scene of a prison at night.

Judging art is subjective, thus, we leave many images for the reader to make their own conclusion (Figs. S11,& S12). In overall, we found the result to be two-fold: this method of activating multiple neurons at the same time could be used to 1) generate creative art images for the image generation domain; and 2) uncover new, unique facets that a neuron has learned to detect—a class of *multifaceted feature visualization* introduced in [9] for better understanding DNNs.

Figure S7: In this experiment, we take 5 classes: school bus, cup, irish terrier, tabby cat, and hartebeest, and split each class into two classes: one containing all canonical images, and one containing all other images. We add these 10 classes back to the ImageNet training set, and train a DNN to classify between all 1005 classes. We show here the result of this experiment for 5 pairs of classes, one pair per row. Per row: the *left* two panels, each shows 9 random images from the training set of a class, and the *right* two panels, each shows 9 visualizations for an output neuron. Our method indeed generates canonical visualizations for neurons trained on canonical images. This result shows evidence that our method reflects well the features learned by the neurons. Also, it suggests that if the visualizations of a neuron do not show canonical images, it's likely that the set of features that the neuron has learned to detect are diverse, and not canonical (e.g. closed-up dog fur texture vs. dog with four legs).

Figure S8: When looking at the visualizations for all output neurons, we found that many pairs of images appear to be very similar, sometimes indistinguishable to human eyes such as hartebeest vs impala, or baboon vs macaque. We investigate whether this phenomenon is reflecting the training set images or it is a shortcoming of our method. Here we show 4 pairs of similar classes. For each pair: the top row shows the top 4 training set images that activate the neuron the most, and the bottom row shows synthetic images produced by our method. An activation score is provided below each image. The result shows that the preferred images closely reflect the training set images that the neurons are trained to classify. For some cases, the difference between two classes can be noticed from both real and synthetic images: *bullfrog* often has a darker, rough skin compared to *treefrog*; *impala* has longer horns and more yellow skin than *hartebeest*. For other cases, it is almost indistinguishable in both real and synthetic images: *Indian* vs. *African* elephant. We also attempted to produce visualizations of more discriminative features by optimizing in the softmax probability output layer instead of the activation (i.e. layer fc8) to make sure that the visualizations of similar classes are different. Preliminary result shows that we indeed obtain distinctive patterns (e.g. between Indian vs African elephant); however, future work is still required to fully interpret them (data not shown).

mask

brambling

leaf beetle

lipstick

bell pepper

beacon

joystick

table lamp

swimming trunks

10%        25%        100%

Figure S9: Visualizations for the output neurons of three DNNs that are trained on 10% (left), 25% (middle) and 100% (right) of ImageNet images respectively. The training and validation accuracy scores of these DNNs are respectively: DNN 10 (100% and 20.2%), DNN 25 (100% and 31.5%), and DNN 100 (79.5% and 57.4%). For each neuron we show 3 images, each starts from different random initializations. This result shows that the image quality somewhat reflects the generalization ability of the net as the images for the overfit DNNs (left two columns) are worse than those for the well-trained DNN (right column). And that our learned prior does not overfit too much to a well-trained encoder DNN because the visualizations for DNN 10 and DNN 25 are still recognizable. All DNNs are given the same number of training updates.

(a) Visualizations of the top 15 output neurons that have the highest class accuracy scores.

(b) Visualizations of the bottom 15 output neurons that have the lowest class accuracy scores.

Figure S10: Visualizations for the output neurons of the classes that the DNN obtains the highest (top) and lowest (bottom) class accuracy scores. For each neuron we show a $2 \times 2$ montage of 4 images, each starts from different random initializations. Below each montage is the class label and class accuracy score. The visualization quality of the neurons with the lowest scores still look qualitatively as good as the neurons with highest scores. This suggests that the low accuracy scores are the result of DNN being confused by pairs of very similar classes in the dataset (e.g. tiger cat vs tiger).

Figure S11: Visualizations of optimizing an image that activates two neurons at the same time. Top panel: the visualizations of activating single neurons. Bottom panel: the visualizations of activating "brambling neuron" and a corresponding neuron shown in the top panel. We found many combinations that are interesting both artistically and scientifically. In other words, this method can be a novel way for generating art images for the image generation domain, and also can be used to uncover new types of preferred images for a neuron, shedding more light into what it does (here are ~30 images that activate the same "brambling" neuron).

Figure S12: Visualizations of optimizing an image that activates two neurons at the same time. Top panel: the visualizations of activating single neurons. Bottom panel: the visualizations of activating "candles" neuron and a corresponding neuron shown in the top panel. In other words, this method can be a novel way for generating art images for the image generation domain, and also can be used to uncover new types of preferred images for a neuron, shedding more light into what it does (here are ∼30 images that activate the same "candles" neuron).

(a) Prior trained to invert conv3 representations.

(b) Prior trained to invert conv5 representations.

(c) Prior trained to invert fc6 representations.

(d) Prior trained to invert fc7 representations.

Figure S13: Comparison of the results of running our optimization framework with different priors $G$, each trained to invert from a different layer of an encoder network $E$. Priors trained to invert features from fully-connected layers (c, d) of an encoder CaffeNet produces better coherent global structures than priors trained to invert from convolutional layers (a, b). These networks are downloaded from [11]. A hypothesis is that each neuron in a lower convolutional layer has a small receptive field size on the input image, and only learns to detect low-level features [9, 2]. Thus, optimizing a code at a convolutional layer results in repeated fragments compared to optimizing at a fully-connected layer, where neurons have learned to care more about global structures [9]. Another orthogonal hypothesis is that the code space at a convolutional layer is much more high-dimensional, making it harder to find codes that produces realistic-looking images.

Figure S14: Visualization of example neuron feature detectors from all eight layers of a CaffeNet DNN [20]. The images reflect the true sizes of the receptive fields at different layers. For each neuron, we show 4 different visualizations: the top 2 images are from a previous work that harnesses a hand-designed prior called "mean image initialization" [9]; and the bottom 2 images are from our method. This side-by-side comparison shows that both method often agree on the features that a neuron has learned to detect. In overall, our method produces more realistic-looking color and texture. However, the comparison also suggests that our method does not visualize well animal faces (the 3rd & 4th units on layer 6; 1st unit on layer 5; and 6th unit on layer 4). Best viewed electronically, in color, with zoom.

Figure S15: Visualization of example neuron feature detectors from all eight layers of a AlexNet DNN trained to classify 205 categories of places from [28]. The images reflect the true sizes of the receptive fields at different layers. For each neuron, we show 4 different visualizations. Similarly to the results from [28], our method also reveals that hidden neurons from layer $3 - 5$ learn to detect objects automatically as the result of training the DNN to classify images of scenes. For example, the 3rd and 4th unit on layer 5 fires for water towers, and fountains respectively. Interesting, we also found that neurons in layer fc6 and fc7 appear to blend different objects together—a similar finding in a different DNN that is trained on ImageNet dataset [9], and also shown in Fig. S14. Best viewed electronically, in color, with zoom.

lighthouse  (9)

building  (56)

sand  (159)

person  (49)

screen  (106)

food  (182)

painting  (46)

bookcase  (155)

swimming pool (96)

water tower (28)

chair  (85)

cementery  (145)

Figure S16: Visualizations of hidden neurons (one per row) in layer 5 of a DNN trained to classify 205 categories of places from [28]. Per row: each of the *left* 5 images (taken from [28]) highlights the region that causes the high neural activation from a real image. This highlighted region is also given a semantic label (shown below each row) by humans from the study in [28]. The *right* 5 images are the visualizations from our method, each produced with a different random initialization. Our method leads to the same conclusions on what a neuron has learned to detect as the method of Zhou et al. [28].

Figure S17: Comparing all previous activation maximization results to the method proposed in this paper (i). For a fair comparison, the categories were not cherry-picked to showcase our best images, but instead were selected based on the images available in previous papers [5, 7–9, 6]. Overall, while it is a subjective judgement and readers can decide for themselves, we believe that our method of synthesizing preferred images via an image generator network prior produces recognizable images with more natural colors and more realistic global structure.

Figure S18: For a better and fair evaluation of our method, we show here 120 *randomly chosen* visualizations for the output neurons of CaffeNet DNN (the model that we use throughout this paper).

(a) Regular ImageNet training images

(b) Cut-up ImageNet training images

(c) Visualizations of neurons that are trained on *regular* ImageNet images

(d) Visualizations of neurons that are trained on *cut-up* ImageNet images

Figure S19: Comparison of the visualizations for neurons that are trained on regular ImageNet images vs. neurons trained on cut-up images. Our result shows evidence that the learned prior is not so strong that it always generates beautiful images. Instead, the visualizations seem to reflect closely the features learned by the neurons, here the features of cut-up objects.

(a) Regular ImageNet training images

(b) ImageNet training images converted into BRG color space

(c) Visualizations of the neurons that are trained on *regular* ImageNet images

(d) Visualizations of the neurons that are trained on *BRG* ImageNet images

Figure S20: Comparison of the visualizations for the neurons that are trained on regular ImageNet images vs. neurons trained on BRG images. Our result shows evidence that the learned prior is not so strong that it always generates beautiful images. Instead, the visualizations seem to reflect closely the features learned by the neurons, here the features of images in a completely different color space.

(a) Regular ImageNet training images

(b) ImageNet training images that are blurred via Gaussian blur (radius=3)

(c) Visualizations of the neurons that are trained on *regular* ImageNet images

(d) Visualizations of the neurons that are trained on *blurred* ImageNet images

Figure S21: Comparison of the visualizations for the neurons that are trained on regular ImageNet images vs. neurons trained on blurred images. Our result shows evidence that the learned prior is not so strong that it always generates beautiful images. Instead, the visualizations seem to reflect closely the features learned by the neurons, here the visualizations of the "blurred" neurons often do not have sharp textures, and the fine details are washed out (e.g. "cardoon" neuron). Best viewed with zoom.

Figure S22: To test whether our method memorizes the training set images, we retrieved the closest images from the training set for each of sample synthetic images. Specifically, for each synthetic image (leftmost column) for an output neuron $Y$ (e.g. lipstick), we find an image among all images of the same class $Y$ with the lowest Euclidean distance in *pixel* space (2nd leftmost), as done in previous works [17], but also in each of the 8 *code* spaces of the encoder DNN (layer conv1 to fc8). The synthetic images are the result of optimizing the input fc6 code of the DGN prior. While comparing against a nearest neighbor among the same class is a much harder test than comparing against a nearest neighbor among the entire dataset, we found no evidence that our method memorizes the training set images. We believe evaluating similarity in the code spaces of deep representations, which better capture semantic aspects of images, is a more informative approach compared to evaluating only in the pixel space.