[Reviews · NeurIPS 2016]

Reviewer 1

Summary

This paper considers the problem of how to synthesise an image to maximise the activation of a given neuron in a feedforward, discriminative neural net. This task is currently a canonical method for visualising the computation being performed in these nets, and is typically achieved by doing gradient descent in image space. To date, the quality of synthesised images have been severely limited by the lack of a good natural image prior. This paper elegantly solves that problem by coupling a pre-trained generative model for images to the discriminative net of interest, and then doing gradient descent within the latent space of the generative model. The authors then systematically study how important it is to "match" the generative model to the discriminative one.

Qualitative Assessment

This is an excellent paper, the best I've reviewed this round. The synthesised images look fantastic, and the analysis seems thorough, honest, and well-motivated. Publish without question. My only criticism for the authors is that they underemphasise the importance of the Dosovitskiy & Brox construction [ref 11] in understanding the rest of the paper. Figure 2 is a great exposition of the model /once G has been fixed/. The statement that G is "trained with principles of GANs" (line 92) gives the impression that the reader is free to ignore the detail to follow, and assume it's just a GAN. Yet the relationship between G and E is the crux of much of the Results. If it's possible to squeeze elsewhere, I would suggest expanding the intro to the description.

Confidence in this Review

3-Expert (read the paper in detail, know the area, quite certain of my opinion)


Reviewer 2

Summary

The authors adopted a pretrained GAN network as prior for visualizing preferred inputs for deep neurons.

Qualitative Assessment

Visual quality of preferred inputs obtained using the proposed method is significantly improved compared to existing methods. Generalizability of the method is also supported by extra experiments. Overall the paper should be interesting to the deep learning community. However, there're also many key questions left untouched. For example, is visual interpretability a sufficient and necessary condition for good network performance? How exactly can we use visual interpretability to improve a network? Also, what are the differences in neural responses induced by natural images vs. preferred inputs generated by existing/proposed methods? The main text also cites supplementary materials very frequently, which might not be very convenient for readers. The authors may consider rearrange their manuscript (e.g. moving details of methods into supplementary materials). %% After Rebuttal %% (No score change.) As most of the reviewers (including myself) agreed that this paper is at least poster worthy for its comprehensive experiments & visually impressive results, I do however want to discuss about the general direction/motivation of this work, corresponding to the authors' response -- "we found no correlation between the activation of a neuron and the recognizability of its visualization" (also in main text): A deep CNN is essentially a nonlinear function that maps an input image to a real value (a target neuron being studied), and activation maximization (AM) is generally considered useful since it tries to discover what images drive a target neuron the most, and those images can be interpreted as "the meaning(s)" of the neuron. However, if like the authors said, recognizability has no correlation against the functional value (activation) of the function (DCNN) being studied, how can we be convinced that better visual recognizability (this work) is a substantial improvement to existing AM methods in helping us better understand these functions? Put another way, how much more understanding about DCNNs have we gained using the proposed method, compared to using methods from e.g. Simonyan et al., whose results are described as uninterpretable (but may actually induce higher activations, since less regularization on the input space is imposed -- unfortunately one of my questions not answered by the authors)? My arguments are in no way trying to reject this paper, but hopefully to help the authors reevaluate their view on AM/preferred inputs and on toward understanding DCNNs. That is, preferred inputs (alone) does not describe/characterize a neuron very well. For example, (Machens et al.) gave a nice example of the efficient coding hypothesis (Barlow; Simoncelli) that the (most) preferred inputs of a grasshopper's auditory neuron would be something completely out of its natural environment, and what really matters should be how well the neuron's tuning curve matches the statistics of natural inputs. Similarly, preferred inputs of DCNN neurons may not be revealing very much about those neurons, even though we feel visually they are. I hope this example can help the authors further improve their paper, especially the discussion/future direction section (but up to the authors' discretion). Machens et al., "Testing the efficiency of sensory coding with optimal stimulus ensembles," Neuron, 2005 Barlow, "Possible principles underlying the transformations of sensory messages," Sensory Communication, 1961 Simoncelli, "Vision and the statistics of the visual environment," Current Opinion in Neurobiology, 2003

Confidence in this Review

3-Expert (read the paper in detail, know the area, quite certain of my opinion)


Reviewer 3

Summary

The paper 'Synthesizing the preferred inputs for neurons in neural networks via deep generator networks' proposes a method to generate images that look close to natural and that maximise the activation of units in a deep network. The method uses a generator network from earlier, related work that was trained to reconstruct images from deep network feature representations. It uses a gradient procedure to find an input to the generator network, which maximises the activation of a unit in the deep network. Hence the generator network acts as an image prior because it constraints the images that can be synthesised during the optimisation. The manuscript describes two main possible applications for the technique: 1. Analysis and visualisation of the feature representations learned by a deep network 2. Synthesis of natural images with controlled semantic properties. Showing extensive results for each of them.

Qualitative Assessment

I see the main contribution presented in the manuscript to be a very intuitive analysis tool for deep neural networks. Studying the results I felt like gaining several interesting insights. For example that images generated from a specific unit during learning often look like they are oscillating between different 'high activation' regions. It could also potentially be a strong tool for semantic image synthesis. However, in the current version of the manuscript it is not clear to what extend it only retrieves images from the training set and small variations thereof. The usefulness of this work will very much depend on how well the relationship between training set and image generation will be understood. Nevertheless, I believe that the results presented in this work are of interest for the deep learning / image generation community and will probably be widely applied or built upon.

Confidence in this Review

2-Confident (read it all; understood it all reasonably well)


Reviewer 4

Summary

The paper proposed a novel method to synthesize preferred inputs for neurons. The method is based on activation maximization. Different from previous works, it utilizes pre-trained generator network in GAN to produce more human-interpretable images. The experiments also validate the effectiveness of the method in many aspects.

Qualitative Assessment

The paper proposed a reasonable way to visualize the preferred inputs for neurons. However, it is still an incremental work on previous methods including activation maximization and deep generator networks, thus leading to the sub-standard novelty. Additionally, the visualization performance of the method for GoogleNet and ResNet which do not have the same architecture as E network are not so good. This limits the usage of the proposed method, as one has to retrain the G network following [11] for different architectures to obtain the best performance. In this case, one can use the G and E network in GAN [22] to update the image and visualize the preferred inputs directly and does not need to resort to other techniques. This lowers the value of the proposed method.

Confidence in this Review

2-Confident (read it all; understood it all reasonably well)


Reviewer 5

Summary

This paper improves image quality of activation maximization by leveraging a generative network as an image prior. They apply their visualization technique to synthesize preferred images for different classes and neurons and produce noticeably improved synthetic images.

Qualitative Assessment

Overall, I thought this was a good paper and was amazed by the quality of the synthetic images. Conceptually, this work is an incremental improvement over prior work, using a complex learned prior instead of hand-designed fixed prior. But the quality of the visualizations are drastically improved, and the utility of this technique is backed up by a large number of experiments. Major concerns: - This technique for activation maximization is specific to continuous datasets, and requires access to a complicated learned image prior over a continuous space. To deploy it in a novel domain, one would first need to train a large differentiable generative model, which we only currently have for images. - While the visualizations are certainly more natural, it's not clear that we learn more from them over existing techniques. What's the benefit of this technique over simply choosing images from the dataset that maximize the activity of a particular neuron? Minor concerns: - [11] also contains VAE-style models that impose a prior over latent space. Why didn't you use these models? It would allow you to avoid the heuristic clipping of the latent code. - Very subjective, but the visualizations for intermediate layers in e.g. S14 look equally uninterpretable between the two techniques. One major goal of all these visualization techniques is understanding these intermediate layers, and there's no evidence that this paper has made improvements here. - Limited generalizability to new architectures and datasets. Some of the preliminary experiments point to a degradation in quality when used with GoogleNet or when trained on places instead of imagenet. Does this technique still work for drastically different architectures like deep resnets?

Confidence in this Review

2-Confident (read it all; understood it all reasonably well)